# Biological Characterization and Genomic Analysis of Three Novel *Serratia*- and *Enterobacter*-Specific Virulent Phages

**DOI:** 10.3390/ijms25115944

**Published:** 2024-05-29

**Authors:** Dziyana Shymialevich, Stanisław Błażejak, Paulina Średnicka, Hanna Cieślak, Agnieszka Ostrowska, Barbara Sokołowska, Michał Wójcicki

**Affiliations:** 1Culture Collection of Industrial Microorganisms—Microbiological Resources Center, Department of Microbiology, Prof. Wacław Dąbrowski Institute of Agricultural and Food Biotechnology—State Research Institute, Rakowiecka 36 Str., 02-532 Warsaw, Poland; diana.szymielewicz@ibprs.pl (D.S.); hanna.cieslak@ibprs.pl (H.C.); 2Department of Biotechnology and Food Microbiology, Institute of Food Sciences, Warsaw University of Life Sciences (WULS–SGGW), Nowoursynowska 166 Str., 02-776 Warsaw, Poland; stanislaw_blazejak@sggw.edu.pl; 3Department of Microbiology, Prof. Wacław Dąbrowski Institute of Agricultural and Food Biotechnology—State Research Institute, Rakowiecka 36 Str., 02-532 Warsaw, Poland; paulina.srednicka@ibprs.pl; 4Department of Nanobiotechnology, Institute of Biology, Warsaw University of Life Sciences (WULS–SGGW), Ciszewskiego 8 Str., 02-786 Warsaw, Poland; agnieszka_ostrowska@sggw.edu.pl

**Keywords:** virulent bacteriophages, biological characterization, genomic analysis, functional annotation, unconventional food preservation

## Abstract

Due to the high microbiological contamination of raw food materials and the increase in the incidence of multidrug-resistant bacteria, new methods of ensuring microbiological food safety are being sought. One solution may be to use bacteriophages (so-called phages) as natural bacterial enemies. Therefore, the aim of this study was the biological and genomic characterization of three newly isolated *Serratia*- and *Enterobacter*-specific virulent bacteriophages as potential candidates for food biocontrol. Serratia phage KKP_3708 (vB_Sli-IAFB_3708), Serratia phage KKP_3709 (vB_Sma-IAFB_3709), and Enterobacter phage KKP_3711 (vB_Ecl-IAFB_3711) were isolated from municipal sewage against *Serratia liquefaciens* strain KKP 3654, *Serratia marcescens* strain KKP 3687, and *Enterobacter cloacae* strain KKP 3684, respectively. The effect of phage addition at different multiplicity of infection (MOI) rates on the growth kinetics of the bacterial hosts was determined using a Bioscreen C Pro growth analyzer. The phages retained high activity in a wide temperature range (from −20 °C to 60 °C) and active acidity values (pH from 3 to 12). Based on transmission electron microscopy (TEM) imaging and whole-genome sequencing (WGS), the isolated bacteriophages belong to the tailed bacteriophages from the *Caudoviricetes* class. Genomic analysis revealed that the phages have linear double-stranded DNA of size 40,461 bp (Serratia phage KKP_3708), 67,890 bp (Serratia phage KKP_3709), and 113,711 bp (Enterobacter phage KKP_3711). No virulence, toxins, or antibiotic resistance genes were detected in the phage genomes. The lack of lysogenic markers indicates that all three bacteriophages may be potential candidates for food biocontrol.

## 1. Introduction

Consuming a diet that includes a variety of fruits and vegetables is crucial for maintaining good health, reducing the risk of diseases, and promoting overall well-being. For this reason, there has been an increase in demand for minimally processed food products [1,2,3]. The production of minimally processed food is limited to basic procedures (washing, peeling, packaging), which allows the obtaining of a ready-to-eat product with a short shelf life (about 4–7 days) [4,5]. The environment of raw plant materials promotes faster growth of microorganisms, which leads to the production of undesirable chemical compounds, reduces the nutritional value, and may become harmful to the consumer’s health [6,7]. Recent reports from the EFSA and ECDC have shown an increasing trend in the number of cases and epidemiological outbreaks caused by the consumption of contaminated raw plant materials [8]. Recently, fresh vegetables have been indicated as an important reservoir of *Enterobacteriaceae*-producing extended-spectrum β-lactamase (ESBL) and AmpC, which raises concerns [9,10]. The spread of β-lactamase-producing bacteria in the food chain may lead to the acquisition of resistance to this type of antibiotic, which is associated with limited therapeutic options and the need to implement/search for innovative treatment methods [11].

Bacteria from the *Serratia* genus belong to the *Yersiniaceae* family and *Enterobacterales* order. *Serratia* are Gram-negative, facultatively anaerobic, non-spore-forming, straight rod-shaped bacteria [12,13]. *Serratia liquefaciens* and *Serratia marcescens* are widespread and occur in soil, water, and the digestive tract of insects, fish, rodents, and humans [14]. They can grow at low temperatures and are capable of secreting thermostable proteases and lipases, which may consequently lead to the production of biogenic amines (e.g., cadaverine and putrescine) [15,16,17]. *S. liquefaciens* and *S. marcescens* often cause spoilage of meat products and frozen vegetables [18,19]. These species can also cause food poisoning or cause illness (gastroenteritis) if they occur in food products [20,21]. A relatively low number of *S. marcescens* (even 10–100 CFU) can be enough to cause an infection; however, the infectious dose may vary depending on the infected individual [20]. *S. marcescens* easily adapts to changing physicochemical conditions, thus demonstrating the ability to survive in various environments, including disinfection solutions, which makes it one of the important nosocomial pathogens [22,23].

Bacteria from the *Enterobacteriaceae* family (*Enterobacterales* order) are considered indicator bacteria of the microbiological quality of food and the hygienic production process [24,25]. This family includes, among others, rods of the *Enterobacter* genus. *Enterobacter* are Gram-negative, facultatively anaerobic, non-spore-forming, rod-shaped bacteria. They constitute part of the intestinal microbiota in 40–80% of the human population [26,27]. These species were often isolated from food processing plants, meat and vegetable products, and rice [28,29]. *Enterobacter cloacae* can cause opportunistic infections, including infections of the eyes and blood, wounds, soft tissues, skin, pneumonia, and urinary tract infections [30,31]. This species has been reported to be resistant to several antibiotics, such as ampicillin, rifampicin, erythromycin, and sulfamethoxazole [32,33]. Moreover, *S. marcescens* and *E. cloacae* exhibit a natural resistance to some antibacterial drugs, including macrolides and penicillins (ampicillin, amoxicillin, amoxicillin with clavulanic acid) [34].

The difficulty in eliminating *S. liquefaciens*, *S. marcescens* and *E. cloacae* lies in the ability of these bacteria to secrete extracellular polymers, which allows the creation of dense biofilms that protect bacterial cells against environmental factors and hinders their elimination from raw materials and surfaces of food production lines [21,35,36]. A promising technique for extending the shelf life of food is the use of strictly lytic bacteriophages or their enzymes, which do not affect the nutritional value of the product and do not change the sensory characteristics [37,38,39]. Before practical application in biocontrol or phage therapy, a comprehensive characterization of bacteriophages is needed. Therefore, in this study, three bacteriophages specific for saprophytic bacteria from the *Enterobacterales* order were isolated from plant-based food products.

Bacteriophage particles were visualized using transmission electron microscopy (TEM), biological characteristics were determined, and genomic analyses were performed. This study aimed to enrich the Culture Collection of Industrial Microorganisms—Microbiological Resources Center of the Department of Microbiology at the Prof. Wacław Dąbrowski Institute of Agricultural and Food Biotechnology—State Research Institute (IAFB) resources with new phages targeting saprophytic bacteria from the *S. liquefaciens*, *S. marcescens*, and *E. cloacae* species to obtain potential candidates for food biocontrol.

## 2. Results and Discussion

### 2.1. Phage Morphology

Three bacteriophages were isolated from municipal sewage: Serratia phage KKP_3708 (phage titer 2.23 × 10^8^ PFU mL^−1^), Serratia phage KKP_3709 (phage titer 9.41 × 10^9^ PFU mL^−1^), and Enterobacter phage KKP_3711 (phage titer 1.23 × 10^9^ PFU mL^−1^) targeting *Serratia liquefaciens* strain KKP 3654, *Serratia marcescens* strain KKP 3687, and *Enterobacter cloacae* strain KKP 3684, respectively. Bacteriophages formed plaques with a transparent halo zone that indicates the production of enzymes from the group of polysaccharide depolymerases, degrading capsular and structural polysaccharides, including exopolysaccharides (EPS), which are the dominant component of the bacterial biofilm. EPS depolymerases are believed to be effective in dispersing biofilms [40]. Photos of phage plaques on bacterial lawn, the procedure for isolating bacteriophage enzymes, and assessing their impact on limiting bacterial growth were described in an earlier article by Shymialevich et al. [41]. The morphology of the isolated bacteriophages was visualized by TEM, and the electron micrographs are shown in Figure 1.

All three isolated bacteriophages belong to the *Caudoviricetes* class (tailed bacteriophages) and contain an icosahedral head and long non-contractile tails (typical siphovirus morphology). Tailed bacteriophages represent the most widespread and numerous group of bacterial viruses [42].

### 2.2. Bacterial Host Range of Bacteriophages

Twenty-five strains from the *Enterobacterales* order were used to determine the host range of newly isolated phages. This bacterial host range was also checked in the study by Wójcicki and co-workers [39] to determine the effectiveness of a phage cocktail for biocontrol against saprophytic bacteria in ready-to-eat plant-based food [39]. The results obtained for Serratia phage KKP_3708, Serratia phage KKP_3709, Enterobacter phage KKP_3711 and the calculated efficiency of plating (EOP) are presented in Table 1.

Among the tested bacteriophages, Enterobacter phage KKP_3711 targeting *Enterobacter cloacae* strain KKP 3684 (as primary host) demonstrated the widest host range and infected 28% of the tested bacteria (7/25). Based on the calculated EOP, it was found that the *Escherichia coli* strain KKP 3705 enables high production of Enterobacter phage KKP_3711 virions (EOP > 0.5). Serratia phage KKP_3708 targeting *Serratia liquefaciens* strain KKP 3654 (as primary host) infected 13% (3/25) of the tested bacterial hosts. According to the obtained EOP, *Enterobacter cloacae* strain KKP 3692 allows for effective infection and phage production (EOP > 0.5). Other bacterial strains will not enable high efficiency of plating. In the case of Serratia phage KKP_3709 targeting *Serratia marcescens* strain KKP 3687 (as primary host), activity was observed only against one strain (EOP < 0.1) The EOP evaluates the seeding efficiency, i.e., the ability of a bacteriophage to form plaque in a given strain compared to a primary bacterial host [43,44]. According to another division, EOP values can be divided into several categories based on their values. An EOP value below 0.001 indicates that the phage is not effective against the target bacteria or is avirulent. EOP values ranging from 0.001 to 0.099 are considered moderately virulent, EOP values ranging from 0.1 to 1.0 indicate that the bacteriophages are highly virulent, indicating the strong effectiveness of the phage against the target bacteria [45]. In the case of Serratia phage KKP_3708, this phage was classified as highly virulent. Depending on the bacterial host, Serratia phage KKP_3709 was classified as moderately or highly virulent, while Enterobacter phage KKP_3711 as avirulent, moderate, or virulent. The variation in efficiency against target bacteria could be caused by inadequate bacteriophage adsorption into target bacteria. Moreover, the resistance mechanisms of the target bacteria could hinder the bacteriophage replication, leading to a lower EOP value [46].

Assessment of the host range is an important step in the characterization of bacteriophages, which enables the selection of bacterial viruses for the development of a broad-spectrum biopreparation [47]. The phage cocktail also prevents bacteria from developing resistance to phage infection, which results in the effective elimination of undesirable bacterial microbiota [48]. Although bacteriophages are very specific, it has been shown that some bacteriophages can infect bacteria within a complete genus or even a family. As an example, salmophages isolated by Wójcicki and co-workers [38], in addition to *Salmonella enterica* species, also infected saprophytic bacterial strains from *Escherichia coli* and *Enterobacter cloacae* species [38]. Understanding the complex dynamics of bacteria–phage interaction will ensure the development of effective biocontrol and phage therapy [49].

### 2.3. Bacterial Growth Inhibition after Phage Infection

Determining the equation of the dependence of optical density on the concentration of bacterial cells (data unpublish) made it possible to adjust the appropriate values of the infection coefficient (MOI = 1000; MOI = 100; MOI = 10; MOI = 1.0; MOI = 0.1; MOI = 0.01; and MOI = 0.001). The results of the dependence of the change in optical density of the bacterial cultures on the MOI used are graphically presented in Figure 2.

In each case, infection with the tested bacteriophages resulted in a delay in the logarithmic growth phase of the bacterial culture compared to the control culture (bacterial strain not treated with phage). Serratia phage KKP_3708 caused complete inhibition of *Serratia liquefaciens* strain KKP 3654 growth at an MOI of 1.0 and above (Figure 2A). In the case of Serratia phage KKP_3709, the complete elimination of *Serratia liquefaciens* strain KKP 3687 was not achieved at the MOIs used (Figure 2B). However, a reduction in optical density was observed depending on the MOI rates. The use of the lowest MOI values (i.e., 0.001 and 0.01) for the Enterobacter phage KKP_3711 resulted in an initial reduction in optical density, which reached control culture values during incubation. However, MOIs of 1000 and 100 completely limited the growth of *Enterobacter cloacae* strain KKP 3684 (Figure 2C).

By analyzing the minimum inhibitory multiplicity of infection (miMOI) of a given bacteriophage, the most effective MOI value to use during application can be determined. The miMOI is defined as the minimum bacteriophage concentration required to inhibit bacterial growth (host), while MOI itself is the ratio of bacteriophages to its reference host bacteria. Based on bacterial growth curve analyses, the miMOI of Serratia phage KKP_3708 was determined to be 1, while the miMOI of Enterobacter phage KKP_3711 was 100. The miMOI value could not be determined for Serratia phage KKP_3709. Hence, Serratia phage KKP_3708 was considered to be more effective since it required a lower phage concentration to inhibit bacterial growth. The obtained results also indicate that the decrease in bacterial growth happened due to an increase in bacteriophage concentration. According to literature data, the different miMOI values in bacteriophages result from their adsorption rate [50].

Determining bacterial growth curves after phage infection is used to predict its effectiveness during application. The study conducted by Cieślik and co-workers [51] showed that Enterobacter phage Entb_43 and Enterobacter phage Entb_45 significantly inhibited the growth of the *E. cloacae* strain 30345 and *E. cloacae* strain 29796, respectively, at a low infection rate (MOI of 0.001). During incubation, the optical density did not change significantly for bacteriophage-treated samples and did not exceed 0.150 [51]. In a study conducted by Wójcicki and co-workers [52], the effect of the addition of bacteriophages was determined at MOI = 1.0 and MOI = 0.1. For all tested bacterial host strains (i.e., *Citrobacter freundii* strain KKP 3655, *Enterobacter cloacae* strain KKP 3082, *Enterobacter ludwigii* strain KKP 3083, and *Serratia fonticola* strain KKP 3084), the use of bacteriophages (i.e., Citrobacter phage KKP 3664, Enterobacter phage KKP 3262, Enterobacter phage KKP 3263, and Serratia phage KKP 3264, respectively) at a higher MOI more strongly limited host cell division (lower optical density values). At the same time, the use of these two MOI values did not completely limit the growth of bacterial hosts during 24-hour culture [52]. According to a study conducted by Shang and co-workers [53] of Salmonella phage vB_SalP_TR2, the inhibitory effect at MOI = 0.01 and MOI = 0.001 was better than that at MOI = 0.1. Application of this phage to milk resulted in a significant reduction in the number of bacterial cells compared to the phage-free control group (reduction of 1.8 log CFU mL^−1^). In the test of salmophage application on the surface of chicken meat, the maximum reduction (0.9 log CFU mL^−1^) was achieved after 6 h of incubation. However, after 12 and 24 h of incubation, no significant statistical differences in pathogen reduction were observed [53].

Regrowth of bacteria after phage treatment may be related to the development of many defense mechanisms in bacteria that protect against phage infection [54]. The basic defense mechanism is masking the receptor through conformation or creating a shell that limits phage interaction [54]. Often, after a phage infection, a bacterial cell can destroy foreign genetic material. In this case, the phage DNA undergoes endonucleolytic digestion in the cell cytoplasm [55,56]. This process involves the recognition of foreign DNA by restrictionases and its cutting and modification through the access of methyl groups [54,57].

### 2.4. Influence of Temperature and pH on the Phage Stability

The influence of temperature in the range from −20 °C to 80 °C on the phage activity was determined (Figure 3). Serratia phage KKP_3709 and Enterobacter phage KKP_3711 retained high titers after 1 h incubation at temperatures up to 60 °C, while the temperature of 80 °C significantly reduced their activity (Figure 3B,C). In the case of Serratia phage KKP_3708, their high activity was retained up to a temperature of 40 °C, but at temperatures of 70 °C and 80 °C, no bacteriophage plaques were observed (Figure 3A).

The effect of a 1 h incubation of phages in solutions with a pH ranging from 3 to 12 is shown in Figure 4. It was observed that incubation at extreme active acidity values (pH 3 and 12) slightly reduced the titer of the tested *Serratia*- (Figure 4A,B) and *Enterobacter*-specific phages (Figure 4C). Bacteriophages retained high activity over the entire pH range tested.

Biological characteristics are a key element allowing the prediction of phage activity in the product environment and its proper selection. The bacteriophage’s stability in environmental conditions is an individual feature [52]. For example, bacteriophages specific to thermophilic bacteria are usually resistant to high temperatures. Research confirmed that bacteriophage AP45 targeting *Aeribacillus* sp. showed activity after 24 h of incubation at 85 °C [58]. Different results were observed in a study conducted by Shymialevich and co-workers [37]. The lysogenic Alicyclobacillus phage KKP 3916 retained high activity in the temperature range from 4 °C to 30 °C and pH from 3 to 11 [37]. While *A. acidoterrestris* grows in a strongly acidic environment and its optimal growth temperature is 42–53 °C, it may also range 35–55 °C [59,60]. In another study, stability testing for the *Salmonella*-specific bacteriophage vB_SalP_TR2 showed limited activity after incubation for an hour at 70 °C and 80 °C and at extreme pH values (i.e., pH 2–3 and pH 11–12) [53]. Phages targeting saprophytic bacteria characterized in a study conducted by Wójcicki and co-workers [52] retained their activity in a wide range of temperatures (from −20 °C to 50 °C) and active acidity values (pH from 4 to 11). All tested phages retained at least 70% of activity after incubation at 60 °C. After 1-hour incubation at 80 °C, no phage activity was observed against the tested bacterial host strains [52]. Phage stability/activity changes during long-term storage. Studies conducted by Cieślik and co-workers [51] showed that phage lysates retained initial titers even during six months of storage at both −70 °C and 4 °C, while storage at 37 °C resulted in a complete loss of their activity.

### 2.5. Phage Genome Sequencing and Analysis Using Bioinformatics Approaches

The complete genomes of the Serratia phage KKP_3708, Serratia phage KKP_3709, and Enterobacter phage KKP_3711 have been sequenced, annotated, and deposited in the GenBank database under the accession numbers OR067836, OR067833, and PP579741, respectively. Moreover, the newly isolated phages were deposited in the Culture Collection of Industrial Microorganisms—Microbiological Resources Center of the Department of Microbiology at the Prof. Wacław Dąbrowski Institute of Agricultural and Food Biotechnology—State Research Institute (IAFB; Warsaw, Poland).

For the genomic characterization of the bacteriophages, we used the latest guidelines on the taxonomy of bacterial viruses published in January 2023 [61]. Similar to TEM, genome analysis confirmed that all three phages belonged to the tailed complex phages from the *Caudoviricetes* class.

The proteomic tree of Serratia phage KKP_3708 generated by the BIONJ program-based TBLASTX genomic sequence comparisons of other phage genomes deposited in the Virus-Host DB [62] is presented in Figure 5A.

The phylogenetic relationship of Serratia phage KKP_3708 to other described prokaryotic viruses on a genome-wide basis was analyzed using ViPTree v4.0 server [63]. Based on the obtained phylogenetic tree, Serratia phage KKP_3708 was classified as a member from the *Autographiviridae* family, *Studiervirinae* subfamily, and *Przondovirus* genus (Figure 5A). Whole-genome sequencing (WGS) revealed that the Serratia phage KKP_3708 genome consists of 40,461 bp linear double-stranded DNA (dsDNA) with a total G+C content of 52.9% (Figure 5B). The described functional proteins have been divided into several groups depending on their functions: related to lysis, structure, DNA metabolisms and replication, and DNA genome packaging (Figure 5B). Out of the 55 predicted open reading frames (ORFs), 32 ORFs are associated with genes encoding proteins with known functions and 23 ORFs encode hypothetical proteins with unknown functions (Figure 5B and Appendix A).

Structural proteins annotated in the Serratia phage KKP_3708 genome included: tail fiber protein, internal virion protein, tail protein, major head protein, head-tail adaptor, head assembly, host range, and adsorption protein. Among the genes associated with lysis, holine, endolysin (amidase), Rz-like spanin, and internal virion proteins with endolysin domain were found. Holins are small hydrophobic proteins that accumulate in the inner cell membrane and, after oligomerization, form pores [65]. This leads to the activation of peptidoglycan hydrolase (endolysin), which accumulates in the periplasm or cytoplasm and causes the degradation of the cell wall at the end of the lytic cycle [65,66]. Lytic enzymes consist of a catalytic domain, which hydrolyzes specific sites in the peptidoglycan, and a cell wall domain. Based on the catalytic domain, five classes of lytic enzymes are distinguished, i.e., glucosaminidase (cut bonds between murein amino sugars), muramidase, amidase (cut amide bonds between the glycan and the peptide bridge), endopeptidase (cut peptide bridges) and lytic transglycosylase [67,68]. Spanins are phage lysis proteins required to disrupt the outer membrane in the final step of Gram-negative host lysis [69]. Among the lytic proteins, an internal virion protein with endolysin domain was annotated, which forms the inner core of the virion and a channel in the host cell envelope, enabling the introduction of phage DNA [70]. Small and large terminase subunits were also detected in the Serratia phage KKP_3708 genome, which plays an important role in packing the viral genome into a pre-formed empty capsid using an ATP-powered molecular motor [71]. It is hypothesized that the small terminase subunit recognizes and binds to the viral genome substrate, while the large subunit is the main component of the terminase holoenzyme and is required for the cleavage of DNA into single genome lengths [71,72]. Serine/threonine kinase detected in the Serratia phage KKP_3708 genome directs the metabolism of the bacterial host for viral replication [73]. In turn, detected Ocr protein prevents the degradation and modification of phage DNA by inhibiting restriction endonuclease [74,75]. dGTPase inhibitor interacts and inhibits host dGTPase/dgt [76]. Two HNH endonuclease coding regions were detected in the Serratia phage KKP_3708 genome, which plays a key role in phage DNA packaging [77]. Among the genes related to replication and metabolism, the following proteins were annotated in the Serratia phage KKP_3708 genome: RNA polymerase, DNA primase/helicase, DNA polymerase I, and exonuclease. Moreover, no toxin or antibiotic resistance genes were detected in the Serratia phage KKP_3708 genome.

BLASTn similarity searches performed for Serratia phage KKP_3708 and related phages deposited in the GenBank database revealed 89.85% nucleotide similarity with 90% query coverage to Klebsiella phage Amrap (GenBank Acc. No. OQ579031.1), 92.36% nucleotide similarity with 88% query coverage to Klebsiella phage vB_Ko_K26PH128C1 (GenBank Acc. No. OY978813.1), 92.79% nucleotide similarity with 85% query coverage to Klebsiella phage VLCpiA3b (GenBank Acc. No. ON602742.1), 92.47% nucleotide similarity with 85% query coverage to Klebsiella phage P79_1 (GenBank Acc. No. OR256027.1), and 92.45% nucleotide similarity with 85% query coverage to Klebsiella phage KP32_isolate 194 (GenBank Acc. No. NC_047969.1) (Figure 6 and Appendix A).

The nucleotide-based virus overall nucleotide sequence identity between isolated Serratia phage KKP_3708 and its 15 closest relatives was calculated using VIRIDIC v1.1 (Figure 7). It was demonstrated that the highest intergenomic similarity (82.9%) with our phage was to Klebsiella phage Amrap (GenBank Acc. No. OQ579031.1). Moreover, Serratia phage KKP_3708 was compared with other bacterial viruses reported in the PhageAI database. Phage similarity on a 2D scatter plot computationally predicted and rendered through PhageAI v1.0.2 software is shown in the Appendix A.

The proteomic tree of Serratia phage KKP_3709 generated by the BIONJ program-based TBLASTX genomic sequence comparisons of other phage genomes deposited in the Virus-Host DB [62] is presented in Figure 8A. The phylogenetic relationship of Serratia phage KKP_3709 to other described prokaryotic viruses on a genome-wide basis was analyzed using ViPTree v4.0 server [63].

Based on the obtained phylogenetic tree, Serratia phage KKP_3709 was classified as a member from the *Myosmarvirus* genus (Figure 8A). WGS revealed that the Serratia phage KKP_3709 genome consists of 67,890 bp linear dsDNA with a total G+C content of 49.8% (Figure 8B). The described functional proteins have been divided into several groups depending on their functions: related to lysis, structure, DNA metabolisms and replication, and DNA genome packaging (Figure 8B). Out of the 194 predicted ORFs, 64 ORFs are associated with genes encoding proteins with known functions and 130 ORFs encode hypothetical proteins with unknown functions. Among the lytic proteins, endolysin and lytic tail proteins were annotated. Among the proteins responsible for packaging, only a large terminase subunit was found. Structural proteins detected in the Serratia phage KKP_3709 genome included: tail protein, baseplate spike, baseplate wedge subunit, tail fiber protein, virion structural protein, major head protein, head maturation protease, portal protein, tail assembly chaperon and minor head protein. Among the proteins responsible for DNA metabolisms and replication, DNA helicase, DNA polymerase, DNA primase, ABC transporter, thymidylate synthase, and polynucleotide kinase were found (Figure 8B and Appendix A). Importantly, no toxin or antibiotic resistance genes were detected in the Serratia phage KKP_3709 genome.

BLASTn similarity searches performed for Serratia phage KKP_3709 and related phages deposited in GenBank database revealed a 98.36% nucleotide similarity with 97% query coverage to Serratia phage vB_SmaM_Hera (GenBank Acc. No. MW021759.1), 97.32% nucleotide similarity with 95% query coverage to Serratia phage MTx (GenBank Acc. No. NC_048759.1), 82.25% nucleotide similarity with 83% query coverage to Erwinia phage vB_EamM_TropicalSun (GenBank Acc. No. MN013090.1), 83.18% nucleotide similarity with 81% query coverage to Serratia phage MyoSmar (GenBank Acc. No. NC_048800.1), and 82.13% nucleotide similarity with 79% query coverage to Serratia phage SMP (GenBank Acc. No. OP490597.1) (Figure 9 and Appendix A).

The nucleotide-based virus overall nucleotide sequence identity between isolated Serratia phage KKP_3709 and its 15 closest relatives was calculated using VIRIDIC v1.1 (Figure 10). Our phage showed the highest intergenomic similarity (95.7%) to Serratia phage vB_SmaM_Hera (GenBank Acc. No. MW021759.1). Furthermore, Serratia phage KKP_3709 was compared with other bacterial viruses reported in the PhageAI database. Phage similarity on a 2D scatter plot computationally predicted and rendered through PhageAI v1.0.2 software is shown in the Appendix A.

The proteomic tree of Enterobacter phage KKP_3711 generated by the BIONJ program-based TBLASTX genomic sequence comparisons of other phage genomes deposited in the Virus-Host DB [62] is presented in Figure 11A. The phylogenetic relationship of Enterobacter phage KKP_3711 to other described bacterial viruses on a genome-wide basis was analyzed using ViPTree v4.0 server [63].

Based on the obtained phylogenetic tree, the Enterobacter phage KKP_3711 shows the highest similarity to bacteriophages from the *Demerecviridae* family and *Markadamsvirinae* subfamily (Figure 11A). According to the Bacterial Viruses Subcommittee (BVS) of the International Committee on Taxonomy of Viruses (ICTV), two phages are assigned to the same species if their genomes are more than 95% identical, while a genus is described as a cohesive group of viruses sharing a high degree (>70%) of nucleotide identity of the full genome length [80]. Following this and based on the results of the comparative genome sequence analysis performed during this study, we considered that Enterobacter phage KKP_3711 represents a new genus of tailed phages from the *Caudoviricetes* class. Moreover, Serratia phage KKP_3708 could not be classified as any species currently recognized by ICTV, and likely represents a new species of tailed phages of siphovirus morphology from the *Caudoviricetes* class. WGS revealed that the Enterobacter phage KKP_3711 genome consists of 113,711 bp linear dsDNA with a total G+C content of 37.8% (Figure 11B). The described functional proteins have been divided into several groups depending on their functions: related to lysis, structure, DNA metabolisms and replication, and DNA genome packaging. Moreover, unlike the Serratia phage KKP_3708 and Serratia phage KKP_3709 genomes, in the Enterobacter phage KKP_3711 genome 27 tRNA regions were located (Figure 11B). The lack of potential tRNA genes suggests that Serratia phage KKP_3708 and Serratia phage KKP_3709 use their host’s tRNA to synthesize their proteins. The lack of tRNA genes may be due to the compressed structure of phage genomes, in which the sequences of translation-related genes are not identified, which may be related to the exclusion of irrelevant information during replication [81]. Out of the 229 predicted ORFs in Enterobacter phage KKP_3711 genome, 49 ORFs are associated with genes encoding proteins with known functions and 120 ORFs encode hypothetical proteins with unknown functions. Proteins related to the phage structure, such as tail fiber, tail protein, capsid protein, portal protein, and baseplate hub protein were detected in the Enterobacter phage KKP_3711 genome. Highlighted proteins are responsible for the adsorption and specificity of the phage (tail fiber) [82]. Other annotated proteins such as holin and endolysin present lytic properties. Large and small subunits of terminase were identified among genes responsible for DNA genome packing. Among the proteins responsible for metabolism and replication, DNA polymerase, exonucleases, endonuclease, helicases, ribonuclease, ligase, and serine/threonine protein were found (Figure 11B and Appendix A). No antibiotic resistance genes, genes encoding virulence factors, integrases, recombinases, or repressors, which are markers of temperate bacteriophages, were identified in the genomes of all three phages.

BLASTn similarity searches performed for Enterobacter phage KKP_3711 and related phages deposited in GenBank database revealed a 71.54% nucleotide similarity with 11% query coverage to Salmonella phage SP33 (GenBank Acc. No. OR862218.1), 72.05% nucleotide similarity with 9% query coverage to Klebsiella phage vB_Kpn_3 (GenBank Acc. No. MZ079855.1), 71.75% nucleotide similarity with 9% query coverage to Escherichia phage EC142 (GenBank Acc. No. ON185584.1), 71.75% nucleotide similarity with 9% query coverage to Escherichia phage EC104 (GenBank Acc. No. ON185581.1), and 71.75% nucleotide similarity with 9% query coverage to Escherichia phage EC122 (GenBank Acc. No. ON185583.1) (Figure 12 and Appendix A).

The nucleotide-based virus overall nucleotide sequence identity between isolated Enterobacter phage KKP_3711 and its 15 closest relatives was calculated using VIRIDIC v1.1 (Figure 13). It was demonstrated that the highest intergenomic similarity (32.6%) with our phage was to Salmonella phage vB_SalS_ABTNLsp4 (GenBank Acc. No. MW149273.1). Moreover, Enterobacter phage KKP_3711 was compared with other bacterial viruses reported in the PhageAI database. Phage similarity on a 2D scatter plot computationally predicted and rendered through PhageAI v1.0.2 software is shown in the Appendix A.

Based on the genomic analyses performed, it was concluded that all three newly isolated bacteriophages, namely, Serratia phage KKP_3708, Serratia phage KKP_3709, and Enterobacter phage KKP_3711 are virulent and do not encode toxin, antibiotic resistance genes or other lysogenic markers, which makes them potential candidates for food preservation.

## 3. Materials and Methods

### 3.1. Bacterial Host Strains

In this research, twenty-five saprophytic bacterial strains from the *Enterobacterales* order deposited in the Culture Collection of Industrial Microorganisms—Microbiological Resource Center of the Department of Microbiology of the Prof. Wacław Dąbrowski Institute of Agricultural and Food Biotechnology—State Research Institute (IAFB; Warsaw, Poland) were used. All strains were isolated from the plant-based minimally processed food products, i.e., rucola, salad mix with carrot, salad mix of beetroot, washed spinach, and unwashed spinach as part of previously conducted studies. Identification of strains was carried out by amplification of the *16S* rRNA gene region and sequencing was outsourced to Genomed S.A. company (Warsaw, Poland). The raw sequences were analyzed using BLASTn (NCBI) and deposited in the GenBank database.

### 3.2. Isolation of Bacteriophages, Propagation and Purification of Phage Particles

Bacteriophages were isolated from municipal sewage (sewage plant treatment “Mokre Łąki”, Izabelin, Poland). For this purpose, the wastewater sample was centrifuged at 8000 rpm for 10 min (Sorvall LYNX 6000 ultracentrifuge, Thermo Fisher Scientific, Watertown, MA, USA) to separate bacteria and organic debris. The supernatant was then filtered through a 0.45 µm syringe filter (Minisart^®^ NML Cellulose Acetate; Sartorius, Goettingen, Germany). To isolate phages specific to selected bacteria, 20 mL of filtered supernatant, 20 mL of double-concentrated Luria–Bertani broth (composition: 10.0 g L^−1^ of peptone (BLT, Łódź, Poland), 10.0 g L^−1^ of sodium chloride (Chempur, Piekary Śląskie, Poland), and 5.0 g L^−1^ of yeast extract (BLT, Łódź, Poland)) and 1 mL of overnight bacterial culture in Luria–Bertani broth were added to a 50 mL falcon. The prepared culture was incubated at 37 °C for 24 h, then centrifuged at 8000 rpm for 10 min. The supernatant was filtered through a 0.45 µm syringe filter [38]. Phage titers (in PFU mL^−1^) were determined by the double-layer agar plate method [83]. For this purpose, a series of dilutions of the obtained phage lysate were performed. To 500 μL of the corresponding dilution, 100 μL of overnight bacterial culture in Luria–Bertani broth was added. After gentle mixing, the suspensions were allowed to stand at room temperature for 20 min. After incubation, the suspensions were poured onto a nutrient agar plate and poured over 4 mL of liquid-chilled Top agar (Luria–Bertani broth with 0.75% agar-agar, pH 7.0–7.2). The liquid mixture was carefully spilled on the surface of the agar plate and allowed to solidify. After drying, the dishes were stored upside down overnight at 37 °C. After incubation, plaques that formed on the bacterial lawn were computed, considering the dilution factor, and expressed as phage titer.

Single-phage plaques were cut with a scalpel and purified in SM (saline magnesium) buffer [84]. Purification was performed in four rounds of single-plaque passage to ensure that the isolate represented the clonal phage population.

### 3.3. Electron Micrographs of Phage Particles

Transmission electron microscopy (TEM) was used to determine phage morphology according to the protocol described in previous articles [38,52]. Briefly, phage lysates were concentrated and purified by centrifugation at 4 °C and 14,500 rpm for 40 min (centrifuge MiniSpin^®^ plus, Eppendorf, Hamburg, Germany). The supernatant was removed and the pellet was suspended in 2 mL of 100 mM cold ammonium acetate (Chempur, Piekary Śląskie, Poland) filtered through a 0.22 µm syringe filter. The pellet was carefully pipetted and centrifuged again using the previously described parameters. The entire procedure was repeated four times. After the final centrifugation, the pellet was suspended in 50 μL of ammonium acetate. The obtained phage suspension in an amount of 2 µL was applied to carbon-sputtered copper–wolfram mesh grids and left to dry at room temperature. The sample was then stained with 2% uranyl acetate (Warchem, Zakręt, Poland) solution for 1 min. The prepared samples were dried for 12 h under sterile conditions and visualized in a JEM-1220 transmission microscope (JEOL, Tokyo, Japan) at 100,000–200,000× magnification at a voltage of 80 kV.

### 3.4. Determination of the Bacterial Host Range for the Tested Phages

The ability of the strains to be infected by a specific bacteriophage was determined by a drop test. For this purpose, on nutrient agar plates were added 100 µL of overnight bacterial culture and equimoniously spread using 4 mL of 0.75% Top-agar. Then, 5 µL of each lysate of a predetermined titer (in PFU mL^−1^) was applied to the bacterial lawns. Plates were incubated for 24 h at 37 °C. After incubation, the results were interpreted according to the assays: “++”—transparent plaques (sensitive bacterial strain); “+”—cloudy (opaque) plaques; “−”—no plaques (insensitive bacterial strain). The efficiency of plating (EOP) in the case of strains sensitive to phage infection was determined using the formula:(1)$$\mathrm{EOP}=\frac{\mathrm{phage}\;\mathrm{titer}\;\mathrm{on}\;\mathrm{test}\;\mathrm{strain}}{\mathrm{phage}\;\mathrm{titer}\;\mathrm{on}\;\mathrm{host}\;\mathrm{strain}}$$

A series of dilutions with seeding by the double-layer plate method was used to determine phage titers, according to the procedure described earlier. Depending on the average EOP value for a specific phage–bacteria combination, the phages were classified as: EOP ≥ 0.5—“high production”, infection of at least 50% compared to the primary host; EOP ≥ 0.1 but < 0.5—“average production”; EOP > 0.001 but < 0.1 for “low production” efficiency; and EOP ≤ 0.001—“ineffective” [43]. According to another division, phages were classified as: EOP < 0.001—the phage is not effective against the target bacteria or is avirulent; EOP ≥ 0.001 but ≤0.099—moderately virulent phages, while EOP ≥ 0.1 to 1.0—high virulence of bacteriophages, indicating strong effectiveness of the phage against the target bacteria [45]. All experiments were performed in triplicate.

### 3.5. Changes in the Growth Kinetics of Bacterial Hosts after Phage Infection at Different MOIs

Growth curves were performed for each bacterial host (unpublished data). For this purpose, a bacterial strain was inoculated into the Luria–Bertani broth and incubated at 37 °C for 24 h. During the incubation, the dependence of the optical density (DU^®^ 640 spectrophotometer, Beckman Instruments, Inc., Fullerton, CA, USA) on the bacterial concentration [CFU mL^−1^] was performed in triplicate. After plotting bacterial growth curves, the growth kinetics of the bacterial culture after phage infection versus the control was measured using a Bioscreen C Pro automatic growth analyzer (Yo AB Ltd., Growth Curves, Helsinki, Finland). For this purpose, fresh bacterial culture was diluted with Luria–Bertani broth to obtain the desired optical density/number of bacterial cells depending on the phage titer. Then, 180 µL of Luria–Bertani broth, 10 µL of an appropriately diluted bacterial culture, and 10 µL of an appropriately diluted phage lysate were pipetted into each well on a multi-well plate to obtain the desired MOI. MOI values of 1000, 100, 10, 1.0, 0.1, 0.01, 0.001, and 0.0001 were used in this study. Based on the analysis of the growth curves of bacterial hosts, the minimum inhibitory multiplicity of infection (miMOI) of bacteriophages was also determined. All experiments were performed with ten repetitions.

### 3.6. Phage Stability after Exposure to Selected Factors

Bacteriophage suspensions of known concentration were incubated for 1 h at various temperatures (−20 °C, 4 °C, 20 °C, 30 °C, 40 °C, 50 °C, 60 °C, 70 °C, 80 °C) and pH values from 3 to 12. Then, the phage titer was determined using the double-layer plate method as mentioned earlier [85].

### 3.7. Extraction of Bacteriophage Genomic DNA

Phage genomic DNA was isolated using the PureLink^TM^ RNA/DNA Mini Kit (Thermo Fisher Scientific Inc., Carlsbad, CA, USA) according to the manufacturer’s protocol, with modifications by Wójcicki et al. [38]. Briefly, 8 mL of precipitation solution (PEG-NaCl: 2.5 M NaCl, 20% PEG 8000) was added to 40 mL of phage lysates and incubated overnight on ice. After incubation, samples were concentrated by centrifugation at 27,000 rpm for 1.5 h at 4 °C (Sorvall LYNX 6000 ultracentrifuge, Thermo Fisher Scientific, Watertown, MA, USA). The pellet was suspended in 400 μL of lysis buffer (containing 5.6 μg of carrier RNA) and vortexed for 15 s. Then, 50 μL of proteinase K was added and incubated for 1 h at 56 °C with shaking at 900 rpm (ThermoMixer C, Eppendorf, Hamburg, Germany). After incubation, the tubes were briefly vortexed and 300 μL of 100% ice-cold molecular biology-grade ethanol was added. Samples were vortexed and incubated at 20 °C for 5 min. After incubation, the tubes were briefly centrifuged to remove any drops from the inside of the lids. A total of 675 µL of each sample was transferred to the viral spin column and centrifuged for 1 min at 10,000 rpm (centrifuge MiniSpin^®^ plus, Eppendorf, Hamburg, Germany). Columns were transferred to new wash tubes, and 500 µL of wash buffer was added and centrifuged for 1 min at 10,000 rpm. The procedure was repeated triplicate, with the last centrifugation for 3 min at 14,500 rpm. To elute the genetic material, the viral spin column was transferred to new tubes and 20 µL of RNase-free water was added. After a 1 min incubation, the samples were centrifuged twice for 1 min at 14,500 rpm. DNA purity was measured by the Nanodrop ND-1000 Spectrophotometer (Thermo Fisher Scientific, Watertown, MA, USA), and DNA concentration was quantified by a Qubit 4.0 Fluorometer using the Qubit dsDNA BR Assay Kit (Invitrogen, Carlsbad, CA, USA). DNA samples were stored at 4 °C until further processing for WGS analysis.

### 3.8. Genome Sequencing and Bioinformatic Analysis

The isolated genomic phage DNA was sent to genXone SA (Złotniki, Poland) for next-generation sequencing (NGS) in the WGS application. DNA libraries were prepared using Rapid Barcoding Kit reagents (Oxford Nanopore Technologies, Oxford, UK) according to the manufacturer’s protocol. A sequencing depth of at least 50× genome coverage was assumed. NGS sequencing was performed by nanopore technology on the GridION X5 sequencing device (Oxford Nanopore Technologies, Oxford, UK) under the control of MinKnow v22.10.5. Bases were called with Guppy v6.3.8 Basecaller (Oxford Nanopore Technologies, Oxford, UK), followed by barcode demultiplexing, also using Guppy Barcoder v6.3.8 (Oxford Nanopore Technologies, Oxford, UK), generating a.fastq file for each barcode. De novo assembly of genomes was performed in Flye v2.8.1 software [86] and annotation of phage genomes in Phanotate v1.5.0 [87] and PhaGAA software (available online: http://phage.xialab.info/home (accessed on 24 April 2024)) [88]. Proksee software (available online: https://proksee.ca/ (accessed on 24 April 2024)) [64] was used to visualize phage genomes. Viral proteomic trees of phage genomes were calculated by BIONJ based on genomic distance matrixes and mid-point rooted and were represented in the circular view. Branch lengths were log-scaled. The sequence and taxonomic data were based on Virus-Host DB [62]. The trees were generated using the ViPTree v4.0 server [63]. Genome sequence comparison of newly isolated phages with five other related phage genomes exhibiting co-linearity was detected by TBLASTX using FastANI v1.3.3 software [78] and Proksee software [64]. The overall isolated phages nucleotide sequence identity with the other 15 phages deposited in the GenBank database was calculated using VIRIDIC v1.1 (intergenomic distance calculator) [79]. Phage similarity was computationally predicted on a 2D scatter plot and rendered through PhageAI v1.0.2 software [89]. The phage genomes were deposited in the GenBank database.

### 3.9. Statistical Analysis

All the experiments were repeated at least three times. The graphical data presented were statistically analyzed using Graph Prism v8.0.2 (GraphPad Software Inc., San Diego, CA, USA). One-way ANOVA followed by Tukey’s test with a 95% confidence interval (α = 0.05) was used to assess the impact of selected physical and chemical factors on phage stability. Error bars shown in Bioscreen C Pro plots represent the standard error of the mean (±SEM).

## 4. Conclusions

Due to increasing antibiotic resistance and the difficulty in eliminating bacteria that form biofilms, scientists are turning to lytic bacteriophages and their enzymes. Therefore, in this study, we present the biological and genomic characterization of three newly isolated phages, namely, Serratia phage KKP_3708, Serratia phage KKP_3709, and Enterobacter phage KKP_3711. Based on genome-wide analyses, we could assign a taxonomic classification. The newly isolated phages belong to the tailed dsDNA bacteriophages from the *Caudoviricetes* class. Bacteriophages were characterized as strictly lytic, lacking toxin, antibiotic resistance genes, and other lysogenic markers. *Serratia*- and *Enterobacter*-specific phages have broad-spectrum activity and show a tolerance to a wide range of temperatures (from −20 °C to 60 °C) and active acidity values (pH from 3 to 12).

In future research, a reasonable approach would be to use strictly lytic (virulent) bacteriophages in combination with other conventional (chemical and/or physical) food preservation methods, including spraying with solutions of weak organic acids or hot water. The features of our newly isolated and characterized lytic bacteriophages indicate their potential for application in food biocontrol, for example, as one of the factors in hurdle technology.

## Figures and Tables

**Figure 1 ijms-25-05944-f001:**
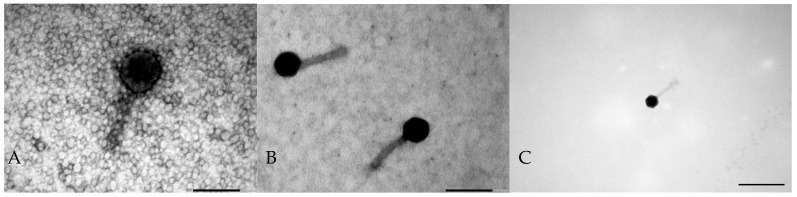
Electron micrographs from the TEM showing the morphology of bacteriophages: (**A**)—Serratia phage KKP_3708; (**B**)—Serratia phage KKP_3709; (**C**)—Enterobacter phage KKP_3711. Scale bar of 200 nm (**A**,**C**) or 100 nm (**B**).

**Figure 2 ijms-25-05944-f002:**
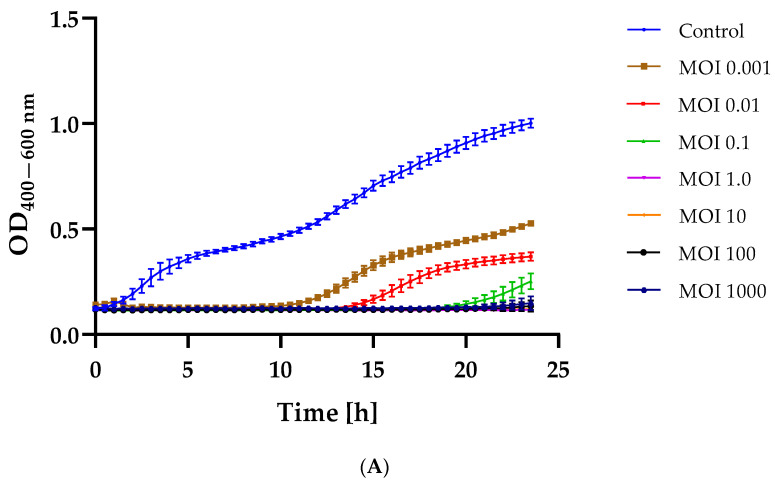
Growth curves of bacterial host strain (*n* = 10) treated with phages at different MOIs compared to the control culture (blue line): (**A**)—*Serratia liquefaciens* strain KKP 3654 infected with Serratia phage KKP_3708; (**B**)—*Serratia marcescens* strain KKP 3687 infected with Serratia phage KKP_3709; (**C**)—*Enterobacter cloacae* strain KKP 3684 infected with Enterobacter phage KKP_3711. Error bars represent the standard deviation (±SD) in optical density values of bacterial cultures.

**Figure 3 ijms-25-05944-f003:**
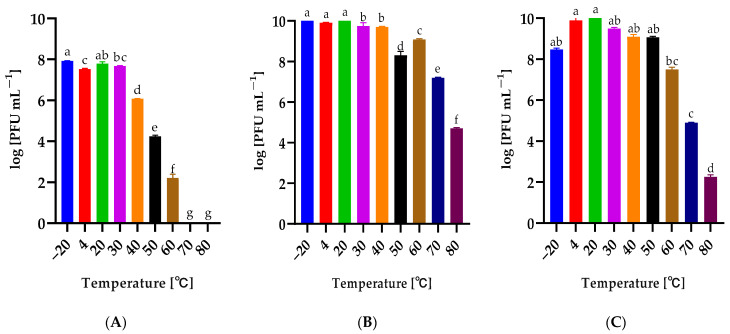
The activity of phages against bacterial host strains after exposure to a wide range of temperatures: (**A**)—Serratia phage KKP_3708, (**B**)—Serratia phage KKP_3709, (**C**)—Enterobacter phage KKP_3711. Letters a, b, c, d, e, f, and g indicate homogeneous groups at a significance level of *p* ≤ 0.05, *n* = 3.

**Figure 4 ijms-25-05944-f004:**
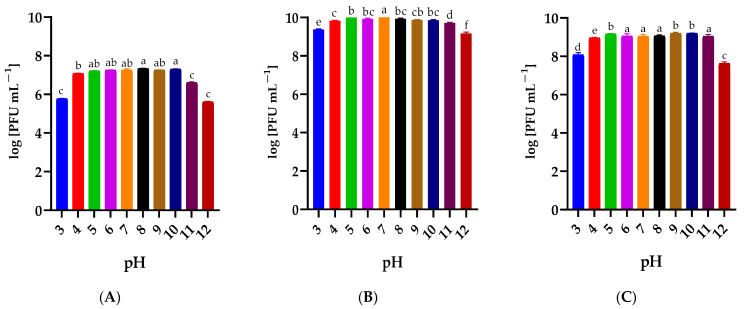
The activity of phages against bacterial host strains after exposure to a wide range of pH values: (**A**) Serratia phage KKP_3708, (**B**) Serratia phage KKP_3709, (**C**) Enterobacter phage KKP_3711. Letters a, b, c, d, e, and f indicate homogeneous groups at a significance level of *p* ≤ 0.05, *n* = 3.

**Figure 5 ijms-25-05944-f005:**
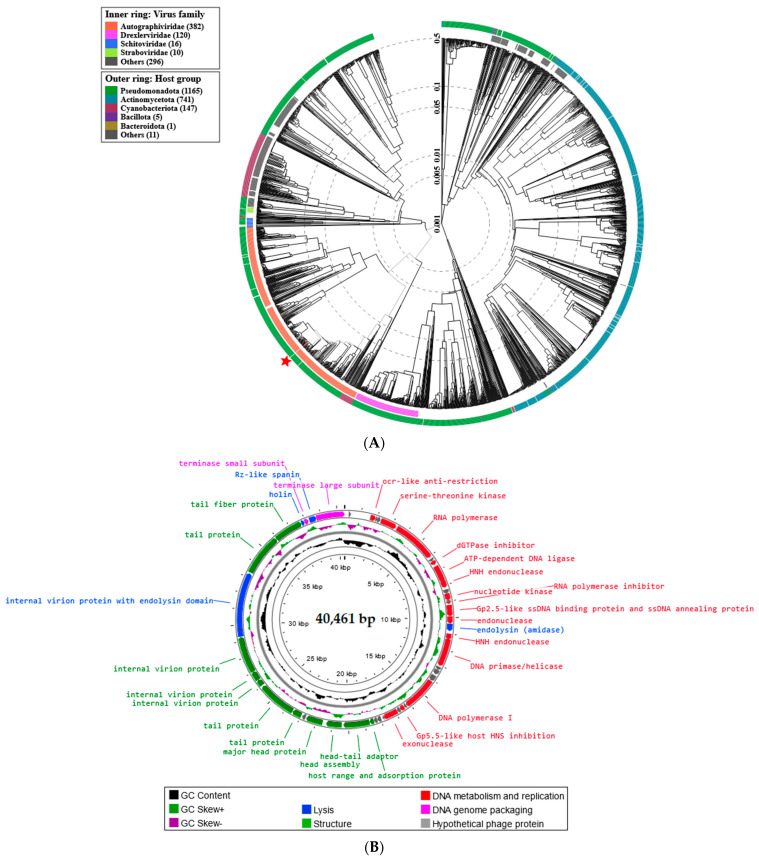
Genome analysis of Serratia phage KKP_3708: (**A**) The viral proteomic tree of Serratia phage KKP_3708 and other phage genomes is shown in a circular view. The branch represented studied phage is marked by an asterisk. Colored rings indicate virus families (inner ring) and host groups (at phylum level; outer ring). The tree was calculated by BIONJ based on genomic distance matrixes and mid-point rooted. Branch lengths are log-scaled. The sequence and taxonomic data were based on Virus-Host DB [62]. The tree shown was generated using the ViPTree v4.0 server [63]. (**B**) Map of the genomic organization of Serratia phage KKP_3708 generated using the Proksee program [64].

**Figure 6 ijms-25-05944-f006:**
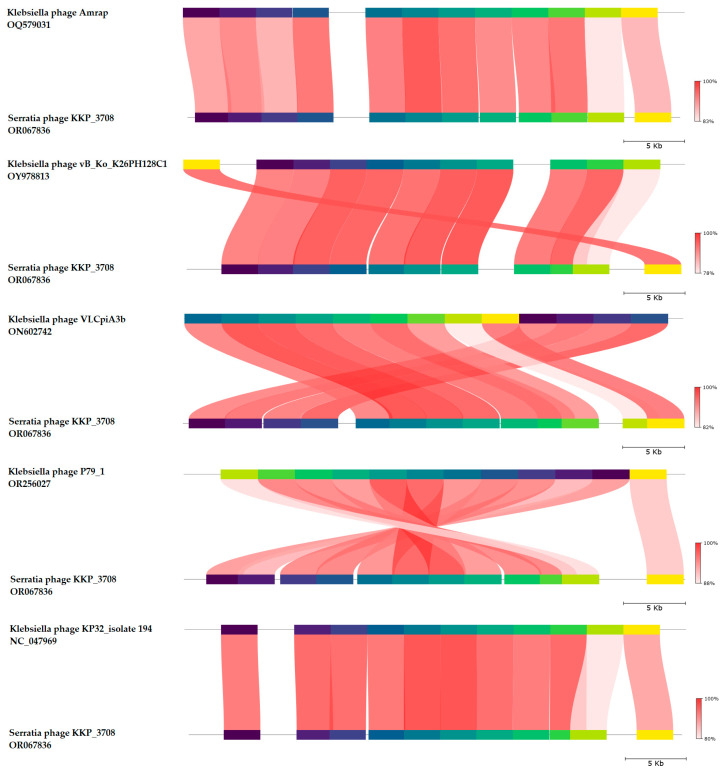
Genome sequence comparison of the Serratia phage KKP_3708 with five other related phage genomes exhibiting co-linearity detected by TBLASTX, using FastANI v1.3.3 software [78] and Proksee software [64]. Homologous regions detected by a TBLASTX search are connected by segments colored based on orthologous matches from query sequence fragments. The color bar shows the percentage identity of TBLASTX.

**Figure 7 ijms-25-05944-f007:**
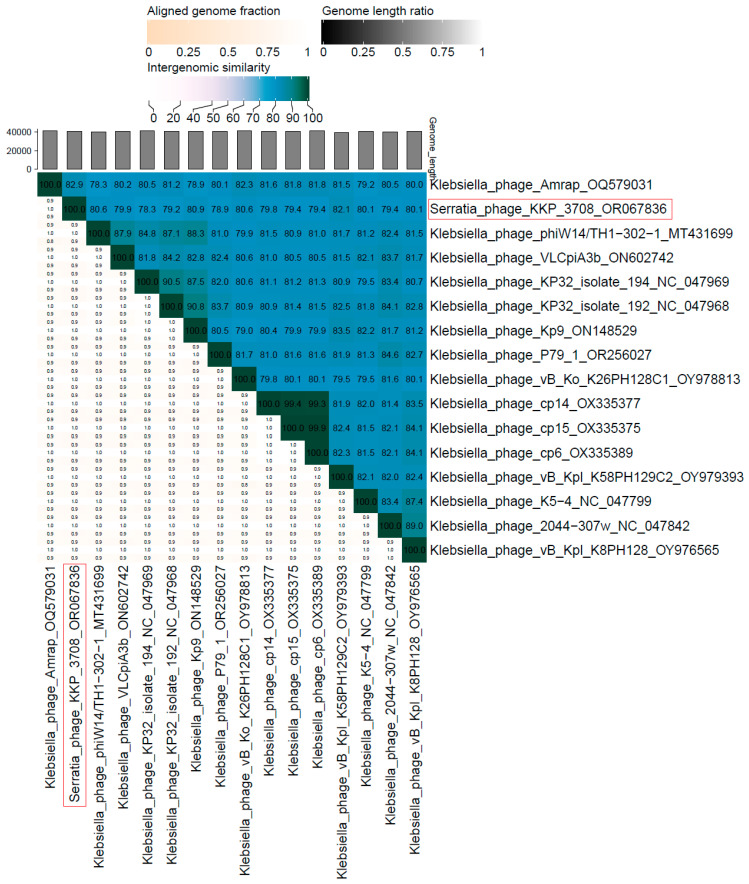
The whole-genome comparison and clustering of Serratia phage KKP_3708 (marked in a red frame), and its 15 closest relatives. The comparison and clustering were performed with the use of VIRIDIC v1.1 [79]. Different shades of blue in the right half of the heatmap represent different intergenomic similarities (%) between the genomes of each pair compared, as indicated above the heatmap and specified by numbers. The left half of the heatmap shows three indicator values for each genome pair: aligned fraction of genome one for the genome in this row (top value), genome length ratio for the two genomes in this pair (middle value), and aligned fraction of genome two for the genome in this column (bottom value).

**Figure 8 ijms-25-05944-f008:**
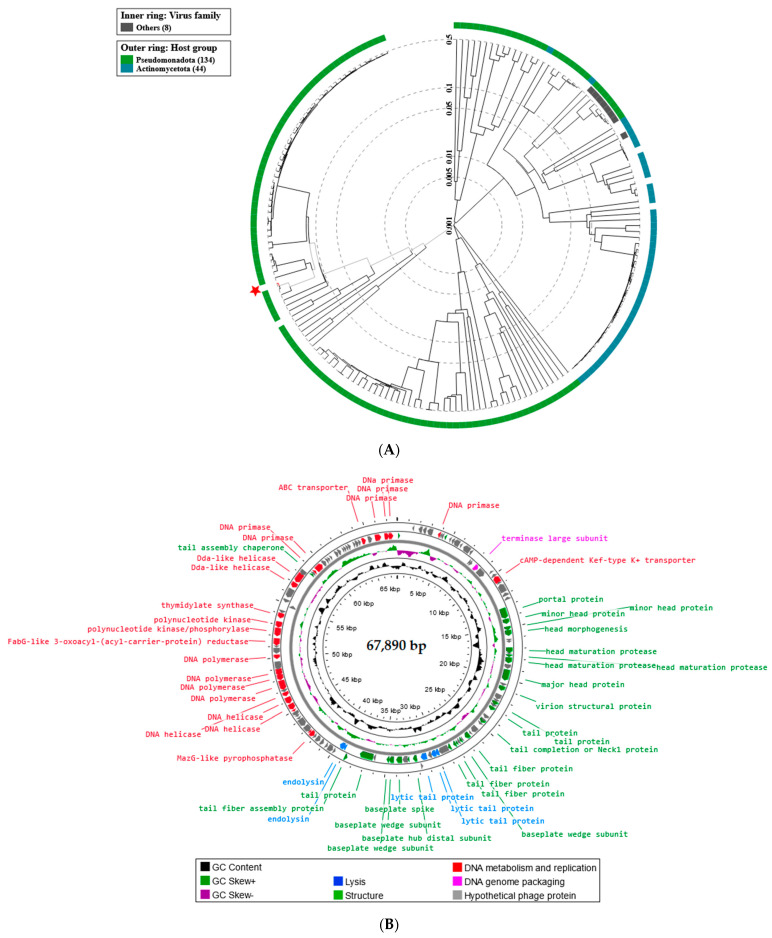
Genome analysis of Serratia phage KKP_3709: (**A**) The viral proteomic tree of Serratia phage KKP_3709 and other phage genomes is shown in a circular view. The branch represented studied phage is marked by an asterisk. Colored rings indicate virus families (inner ring) and host groups (at phylum level; outer ring). The tree was calculated by BIONJ based on genomic distance matrixes and mid-point rooted. Branch lengths are log-scaled. The sequence and taxonomic data were based on Virus-Host DB [62]. The tree shown was generated using the ViPTree v4.0 server [63]. (**B**) Map of the genomic organization of Serratia phage KKP_3709 generated using the Proksee program [64].

**Figure 9 ijms-25-05944-f009:**
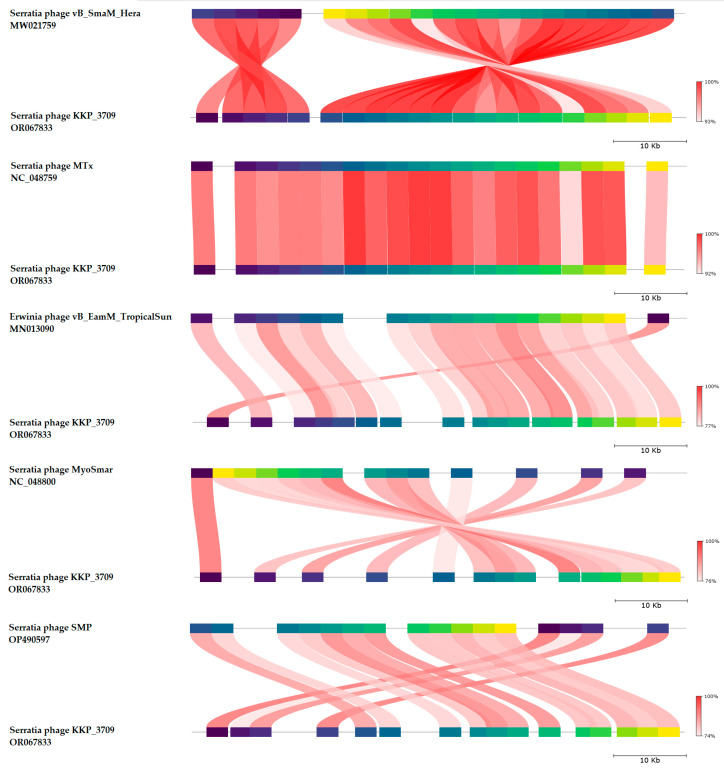
Genome sequence comparison of the Serratia phage KKP_3709 with five other related phage genomes exhibiting co-linearity detected by TBLASTX, using FastANI v1.3.3 software [78] and Proksee software [64]. Homologous regions detected by a TBLASTX search are connected by segments colored based on orthologous matches from query sequence fragments. The color bar shows the percentage identity of TBLASTX.

**Figure 10 ijms-25-05944-f010:**
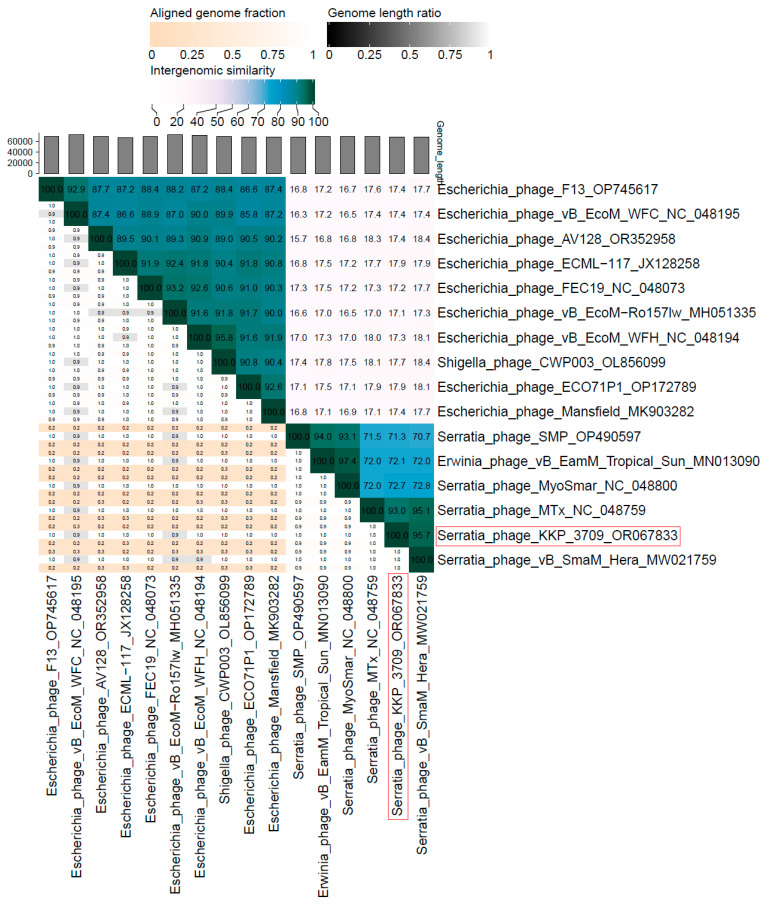
The whole-genome comparison and clustering of Serratia phage KKP_3709 (marked in a red frame) and its 15 closest relatives. The comparison and clustering were performed with the use of VIRIDIC v1.1 [79]. Different shades of blue in the right half of the heatmap represent different intergenomic similarities (%) between the genomes of each pair compared, as indicated above the heatmap and specified by numbers. The left half of the heatmap shows three indicator values for each genome pair: aligned fraction of genome one for the genome in this row (top value), genome length ratio for the two genomes in this pair (middle value), and aligned fraction of genome two for the genome in this column (bottom value).

**Figure 11 ijms-25-05944-f011:**
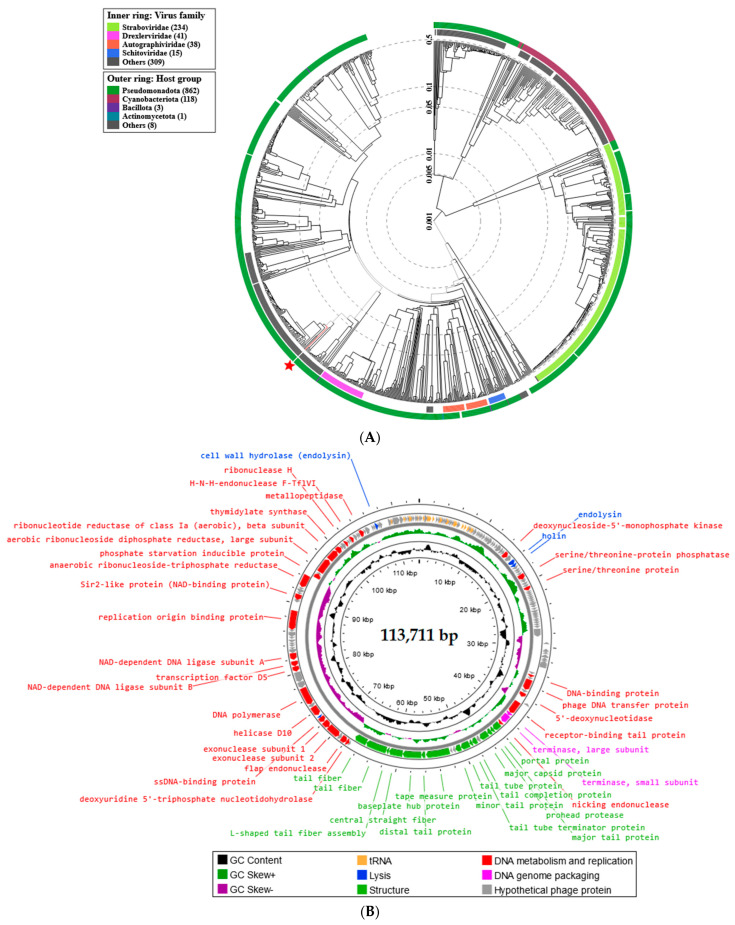
Genome analysis of Enterobacter phage KKP_3711: (**A**) The viral proteomic tree Enterobacter phage KKP_3711 and other phage genomes is shown in a circular view. The branch represented studied phage is marked by an asterisk. Colored rings indicate virus families (inner ring) and host groups (at phylum level; outer ring). The tree was calculated by BIONJ based on genomic distance matrixes and mid-point rooted. Branch lengths are log-scaled. The sequence and taxonomic data were based on Virus-Host DB [62]. The tree shown was generated using the ViPTree v4.0 server [63]. (**B**) Map of the genomic organization of Enterobacter phage KKP_3711 generated using the Proksee program [64].

**Figure 12 ijms-25-05944-f012:**
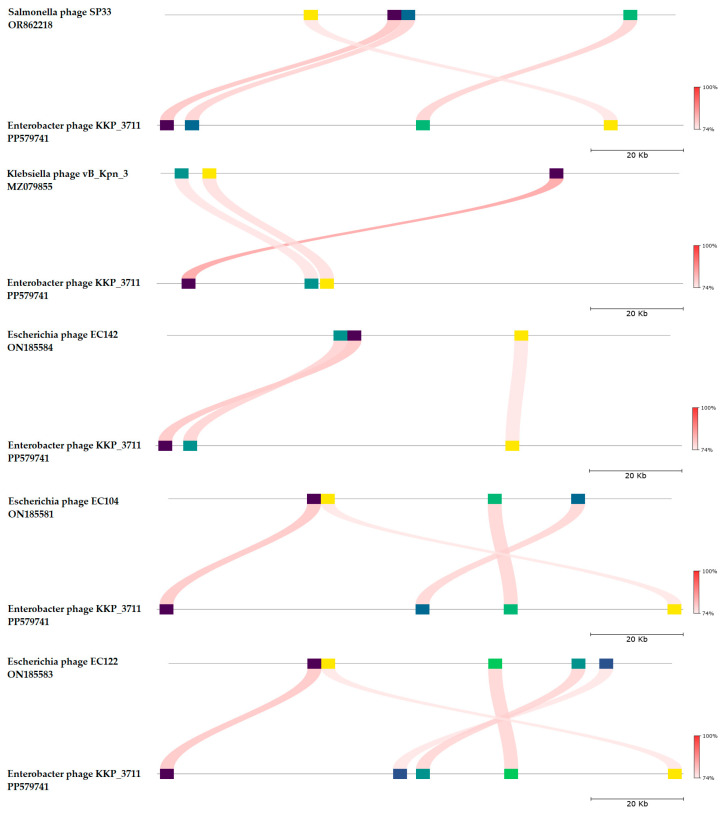
Genome sequence comparison of the Enterobacter phage KKP_3711 with five other related phage genomes exhibiting co-linearity detected by TBLASTX, using FastANI v1.3.3 software [78] and Proksee software [64]. Homologous regions detected by a TBLASTX search are connected by segments colored based on orthologous matches from query sequence fragments. The color bar shows the percentage identity of TBLASTX.

**Figure 13 ijms-25-05944-f013:**
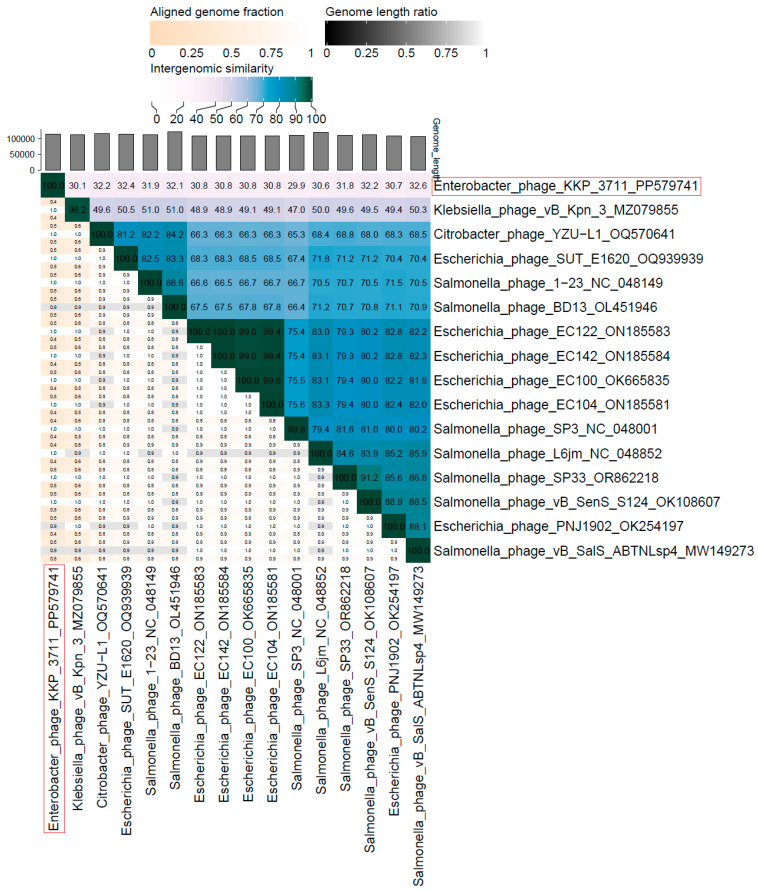
The whole-genome comparison and clustering of Enterobacter phage KKP_3711 (marked in a red frame), and its 15 closest relatives. The comparison and clustering were performed with the use of VIRIDIC v1.1 [79]. Different shades of blue in the right half of the heatmap represent different intergenomic similarities (%) between the genomes of each pair compared, as indicated above the heatmap and specified by numbers. The left half of the heatmap shows three indicator values for each genome pair: aligned fraction of genome one for the genome in this row (top value), genome length ratio for the two genomes in this pair (middle value), and aligned fraction of genome two for the genome in this column (bottom value).

**Table 1 ijms-25-05944-t001:** Range of bacterial hosts for three newly isolated phages.

Bacterial Host Strain (*n* = 25)	GenBank Accession Number	Year of Isolation	Source of Isolation	Phage Strain (EOP)		
				KKP_3708	KKP_3709	KKP_3711
*Enterobacter cloacae* KKP 3082	MZ827006	2018	Salad mix with carrot	−	−	+ (0.215)
*Enterobacter ludwigii* KKP 3083	MZ827002	2018	Salad mix with carrot	−	+ (0.033)	−
*Serratia fonticola* KKP 3084	MZ827668	2018	Salad mix with beetroot	−	−	−
*Escherichia coli* KKP 3650	OM287487	2018	Salad mix with beetroot	−	−	−
*Pantoea agglomerans* KKP 3651	OP978292	2018	Washed spinach	−	−	+ (0.091)
*Serratia fonticola* KKP 3652	OM287486	2018	Washed spinach	−	−	−
*Serratia liquefaciens* KKP 3654	OP978313	2018	Salad mix with beetroot	++ (1)	−	−
*Citrobacter freundii* KKP 3655	MZ827001	2018	Rucola	−	−	−
*Enterobacter cloacae* KKP 3656	OM304355	2018	Unwashed spinach	+ (0.274)	−	−
*Enterobacter cloacae* KKP 3684	OM281790	2018	Salad mix with beetroot	−	−	++ (1)
*Serratia fonticola* KKP 3685	OM281802	2018	Unwashed spinach	−	−	−
*Enterobacter cloacae* KKP 3686	OM281778	2018	Unwashed spinach	−	−	+ (0.305)
*Serratia marcescens* KKP 3687	OK103977	2018	Salad mix with beetroot	−	++ (1)	−
*Escherichia coli* KKP 3688	OM281784	2018	Salad mix with beetroot	−	−	−
*Raoultella terrigena* KKP 3689	OK085529	2018	Salad mix with beetroot	−	−	−
*Escherichia coli* KKP 3691	OM281773	2018	Washed spinach	−	−	−
*Enterobacter cloacae* KKP 3692	OM281803	2018	Unwashed spinach	++ (0.553)	−	+ (0.00003)
*Escherichia coli* KKP 3705	OM212647	2018	Salad mix with beetroot	−	−	+ (0.614)
*Enterobacter cloacae* KKP 3706	OM278533	2018	Salad mix with carrot	−	−	+ (0.024)
*Escherichia coli* KKP 3707	OM281777	2018	Washed spinach	−	−	−
*Escherichia coli* KKP 3800	OM250392	2022	Rucola	−	−	−
*Escherichia coli* KKP 3801	OM250391	2022	Rucola	−	−	−
*Escherichia coli* KKP 3802	OM250393	2022	Washed spinach	−	−	−
*Escherichia coli* KKP 3824	ON303636	2020	Salad mix with carrot	−	−	−
*Escherichia coli* KKP 3825	ON303626	2020	Rucola	−	−	−

Notes: When strain infected with phage: “++”—transparent plaques; “+”—cloudy plaques; “−”—no plaques (insensitive bacterial strain). Each spot test was performed in triplicate (*n* = 3).

## Data Availability

Data are contained within the article and Appendix A.

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
