# Peer review of "Biological Characterization and Genomic Analysis of Three Novel Serratia- and Enterobacter-Specific Virulent Phages"

_ijms, 2024, doi:10.3390/ijms25115944_

Round 1

Reviewer 1 Report

Comments and Suggestions for Authors

REVIEWER REPORT FOR BIOLOICAL CHARACTERIZATION AND GENOMIC ANALYSIS OF THREE NOVEL Serratia– and Enterobacter–SPECIFIC VIRULENT PHAGES

I have read the manuscript titled "Biological Characterization and Genomic Analysis of Three Novel Serratia– and Enterobacter–Specific Virulent Phages". The study aims at the  biological and genomic characterization of three newly isolated Serratia– and Enterobacter–specific virulent bacteriophages as potential strains for food biocontrol. Three strains: Serratia phage KKP_3708 (vB_Sli–IAFB3708), Serratia phage KKP_3709 (vB_Sli–IAFB3709), and Enterobacter phage KKP_3711 (vB_Ecl–IAFB3711) were isolated against Serratia liquefaciens strain KKP 3654, Serratia marcescens strain KKP 3687, and Enterobacter cloacae strain KKP 3684, respectively, thereby assigning a taxonomic classification, The study went ahead to note the effect of phage addition at different multiplicity of infection (MOI) rates on the growth kinetics of the bacterial hosts using a Bioscreen C Pro growth analyzer. The newly isolated phages belong to the tailed dsDNA bacteriophages from the Caudoviricetes class. The bacteriophages were characterized as strictly lytic, lacking toxin, antibiotic resistance genes, and other lysogenic markers. Serratia- and Enterobacter–specific phages have strong lytic activity and show a wide
tolerance of temperatures (from –20 ℃ to 60 ℃) and active acidity values (pH from 3 to 12). The authors conclude that the features of the isolates make them potential candidates for their applications as food biocontrol agents.

The authors of the manuscript has followed the guidelines in writing proposed by the journal, the authors have also given good justification for the work done as stated in the Introduction section; the paper gives good reasons for the need of an innovative method in combating microbiological contamination; citing the problems of antibiotic resistance and why the selected strain is a good fit as a biocontrol agent.
The manuscript is clear and has been presented in a well structured manner, the cited references are relevant and adequate, the design of the work done is appropriate and has been stated in such a way that the results achieved can be reproduced.
The figures and images are clear and represents the data clearly.
Below are my observations:
1. While the abstract is clear, it is not comprehensive; it did not state how the isolation was done. This can be briefly explained in the abstract so that it gives a complete summary of the paper.
2. The introduction seems clearly written, the authors have ensured that the section gives a good overview of the topic, why the topic and the steps taken to ensure that the idea behind the work is completely acceptable.
3. While most of the methods are contained in the paper, there is no section for Methodology, this is necessary to allow for reproducible results based on the details given in this methods section. A section from the results and discussion should be assigned as the Methodology section.
4. The results and discussion section seems clear, there are some parts that could use references to validate the information that has been given;
Lines 226-228
Lines 256-258
Reference number 73 is old and should be replaced.
5.The conclusion section could use more information, deducing that the features of the isolated strain(temperature and pH) makes it ideal as a biocontrol agent doesn't seem sufficient, authors should cite related articles that support this and also explain how other features observed from the research such as strictly lytic, lacking toxin, antibiotic resistance genes, and other lysogenic markers makes for a good biocontrol agent.

While the paper is a good fit and suitable for publication, I recommend a Minor Revision for this paper before publication.

Author Response

Author's Notes in the attachment.

Reviewer 2 Report

Comments and Suggestions for Authors

Very interesting study characterising 3 novel Serratia /Enterobacter phages - authors need to ensure making the sequence data available via GenBank as it is not available for any of the phages; https://www.ebi.ac.uk/ena/browser/view/OR067836

Also, it is very important that authors mention in the discussion section line 165 that: "understanding the complex-dynamics of bacteria-phage interaction will ensure developing effective biocontrol and phage therapy" https://pubmed.ncbi.nlm.nih.gov/37380724/ 

Author Response

Author's Notes in the attachment. 
